# Association between Type 2 Diabetes and Classification of Periodontal Disease Severity in Japanese Men and Women: A Cross-Sectional Study

**DOI:** 10.3390/ijerph19138134

**Published:** 2022-07-02

**Authors:** Nanae Dewake, Yukiko Iwasaki, Akira Taguchi, Nobuyuki Udagawa, Nobuo Yoshinari

**Affiliations:** 1Department of Operative Dentistry, Endodontology and Periodontology, School of Dentistry, Matsumoto Dental University, Shiojiri 399-0781, Japan; nobuo.yoshinari@mdu.ac.jp; 2Department of Oral Sciences, Matsumoto Dental University Hospital, Shiojiri 399-0781, Japan; yukiko.iwasaki@mdu.ac.jp; 3Department of Oral Health Promotion, Graduate School of Oral Medicine, Matsumoto Dental University, Shiojiri 399-0781, Japan; akira.taguchi@mdu.ac.jp (A.T.); nobuyuki.udagawa@mdu.ac.jp (N.U.); 4Department of Oral and Maxillofacial Radiology, School of Dentistry, Matsumoto Dental University, Shiojiri 399-0781, Japan; 5Department of Biochemistry, School of Dentistry, Matsumoto Dental University, Shiojiri 399-0781, Japan

**Keywords:** type 2 diabetes, periodontal disease severity, alveolar bone loss rate, high-sensitivity C-reactive protein

## Abstract

Background: to evaluate the association between type 2 diabetes and periodontal disease severity using the rate of alveolar bone loss (ABL) and high-sensitivity C-reactive protein (hs-CRP) value as indices. Methods: In this cross-sectional study of 372 patients (mean age ± SD, 53.2 ± 11.8 years) from a Japanese hospital, we measured ABL and number of teeth on either panoramic radiographs or intraoral dental radiographs of all teeth. Periodontal disease severity was classified into nine groups by combining ABL and hs-CRP. Results: 48 subjects had type 2 diabetes; 324 did not. Univariate analysis showed that type 2 diabetes was significantly associated with age, sex, body mass index, number of teeth, ABL, hs-CRP, and periodontal disease severity. Multivariate analysis showed significant associations between type 2 diabetes and the groups with high severity of periodontal disease. In receiver operating characteristic (ROC) curve analysis, predicting the presence of diabetes, area under the ROC curve was 0.762 (95%CI = 0.688–0.835) for ABL, and 0.709 (95%CI = 0.635–0.784) for hs-CRP, which was significant. Conclusions: this study showed that diabetes can be associated with a periodontal disease severity classification using the combination of ABL and hs-CRP.

## 1. Introduction

The association between systemic health and oral health is bidirectional; systemic illnesses, especially metabolic disorders, affect oral health, and it appears that oral health may affect systemic health [1]. The presence of periodontal disease often strongly correlates with type 2 diabetes [2,3]. Periodontal disease is a local chronic inflammatory disease, initiated by the accumulation of a pathogenic dental plaque biofilm above and below the gum margin, within which microbial dysbiosis leads to a chronic non-resolving and destructive inflammatory response [4,5]. There is strong evidence that people with periodontitis have an elevated risk for dysglycemia and insulin resistance [6]. Moreover, some cohort studies have demonstrated that patients with type 2 diabetes and periodontitis have significantly higher HbA1c levels compared with patients without periodontitis [6].

In 2019, 1 in 11 adults aged from 20 to 79 years were reported to have diabetes (463 million people) worldwide [7]. In the Western Pacific area, including Japan, the number of people with diabetes is predicted to increase by 31% between 2019 and 2045 [7]. Furthermore, one in two adults with diabetes are undiagnosed globally (232 million) [7]. Although there are several established and accurate screening tools for DM (A1c, fasting glucose, OGTT), it is important to develop an additional diagnostic method to capture at-risk patients in non-traditional clinical setting for detecting type 2 diabetes in the early stages. Periodontal disease is also an asymptomatic disease, and it has the highest prevalence of all infectious diseases. If dentists can predict the early stages of type 2 diabetes from a periodontal examination, it could be effective as a screening tool and reduce medical costs.

Clinical periodontal disease severity has been developed by both the American Association of Periodontology and the European Federation of Periodontology [8,9]. However, the lack of consensus and uniformity in the definition of periodontitis within epidemiological studies is a serious problem [10]. The methods used in the evaluation and analysis of periodontal disease in actual epidemiologic studies are varied, including probing pocket depth (PPD) only and combinations of PPD and bleeding on probing (BOP). The severity classifications of periodontitis established by both the American Association of Periodontology and the European Federation of Periodontology may not be used. The criteria used to evaluate periodontitis in epidemiological studies therefore vary between articles, making it difficult to carry out a standardized evaluation in the meta-analyses and systematic reviews. Therefore, in 2011, the Japanese Society of Periodontology created a trial classification using (ABL) as a clinical index, together with the high-sensitivity C-reactive protein (hs-CRP) value, which is an inflammatory marker [11]. For the clinical evaluation of the new classification, we used the assessment of alveolar bone resorption rate, in combination with high-sensitivity CRP, a biomarker of inflammation common to both medicine and dentistry. We considered periodontal disease to be localized chronic inflammation. The purpose of the new classification was to address the problem of periodontal pocket measurement, which has errors caused by variations in the examiner’s measurements which use probes. The severity of periodontal inflammation can be evaluated using high-sensitivity CRP quantified by biomarkers and alveolar bone resorption rate evaluated by X-ray images, both of which can be objectively measured and evaluated. This study was one of several planned to investigate the association of the new classification with various diseases, and focused on type 2 diabetes. Previous studies showed the association between Hs-CRP and type 2 diabetes [12]. However, little information is available about the association between classifications of periodontal disease severity and the condition of type 2 diabetes.

The purpose of this study was to evaluate the association between type 2 diabetes and the severity of periodontal disease using ABL and the hs-CRP value as indices. Furthermore, we investigated whether ABL and hs-CRP could associate with type 2 diabetes with the aim of using it as a new screening method.

## 2. Material and Methods

### 2.1. Design and Subjects

The design was a cross-sectional study. Participants were 322 patients who had a medical checkup in the medical examination center of Matsumoto Dental University Hospital and 50 patients who visited the department of periodontology at Matsumoto Dental University Hospital from 2012 to 2015 (a total of 372 patients: 252 men and 120 women). The mean age (standard deviation) of the subjects was 53.2 (11.8) years. Of these, 48 were diagnosed with type 2 diabetes by their home doctors and were receiving medication and insulin injection therapy. Age, sex, body height, body weight, and current smoking history were obtained from the medical records of each subject. Before the start of the study, written informed consent was obtained from all subjects for their participation in the study, according to the Declaration of Helsinki.

The following were exclusion criteria for this study. (1) Those who are pregnant or may become pregnant. (2) Those who have uncontrolled severe cardiac disease, renal dysfunction, or hepatic dysfunction; these diseases have been associated with periodontal disease [13,14,15]. (3) Those who are taking antibody drugs or anti-inflammatory drugs for autoimmune diseases. (4) Those who have taken any antibacterial drugs for the past three months from the time of investigation. All subjects were informed of the results of this study in accordance with the ethics guidelines of the Ministry of Health, Labor and Welfare and the Ministry of Education, Culture, Sports, Science and Technology. The Institutional Review Board for Clinical Research at Matsumoto Dental University reviewed and approved this study protocol (no. 0151).

### 2.2. Assessment of ABL Using Oral Radiographs and Hs-CRP

Either panoramic radiographs or intraoral dental radiographs of all teeth were taken during a medical examination or reassessment examination. Panoramic radiographs were taken with a digital AZ3000^®^ device (Asahi Roentgen Ind., Co., Ltd., Kyoto, Japan) and intraoral radiographs were taken with a full-mouth set using DIGORA^®^ Optime (Soredex Orion Corp., Tuusula, Finland).

One periodontist with 5 years of experience examined the radiographs and recorded the number of teeth and the ABL. Implants, supernumerary teeth, and third molars were excluded from the number of teeth. Residual roots without a cap for an overdenture were also excluded. Teeth with caries or periapical lesions were not excluded.

ABL was assessed on a panoramic radiograph or intraoral radiographs [16] by measuring the distance between the cement-enamel junction (CEJ) and the alveolar crest (AC) and between the CEJ and the root apex at two sites (mesial and distal) on each tooth. The apex was defined as the most apically located point of the root. In teeth restored with fillings or crowns, the most apical limit of the restoration was considered equivalent to the CEJ. Finally, ABL was calculated as a CEJ-AC/CEJ-apex [17].

Hs-CRP values were measured from the serum with a Latex agglutination/nephelometry method in the SRL Hachioji Lab (Tokyo, Japan). Blood was collected by a clinical technician in the clinical laboratory in Matsumoto Dental University Hospital. The collected blood was centrifuged (28 °C, 5 min, 3600 rpm) and stored in serum.

### 2.3. Classification of Periodontal Disease by Severity

The classification used in this study is a new trial classification of periodontal disease. The classification is based on the following categories: ABL of less than 25% is clinically mild (I), 25% or more and less than 35% is moderate (II), 35% or more is severe (III). An hs-CRP value of less than 440 ng/mL is mild inflammation (A), 440 ng/mL or more and less than 1020 ng/mL is moderate (B), and 1020 ng/mL or more is severe (C). Combining ABL with hs-CRP yields nine classifications of periodontal disease severity [11] (Figure 1).

### 2.4. Statistical Analysis

Initially, univariate analyses with the t-test and the chi-squared test were used to evaluate the differences in age, sex (binary), body mass index (BMI), current smoking history, number of teeth, ABL (three groups), and hs-CRP value (three groups) and between subjects with and without type 2 diabetes. Next, multivariate logistic regression analysis was undertaken with forward selection adjusting for age, sex (binary), BMI, current smoking history (binary), number of teeth, and the nine classifications of periodontal disease severity. Receiver operating characteristic (ROC) curve analysis was employed to identify asymptomatic type 2 diabetes in relation to ABL and hs-CRP value. According to the method suggested by Swets [18], the area under the ROC curve (AUROC) was determined as follows: less accurate (0.5 < AUROC < 0.7), moderately accurate (0.7 < AUROC < 0.9), highly accurate (0.9 < AUROC < 1), and perfect tests (AUROC = 1). All comparisons were two-sided and performed at a *p* = 0.05 level of significance. Statistical analysis was performed using SPSS^®^ ver. 26.0 for Windows (IBM Japan, Tokyo, Japan).

## 3. Results

The characteristics of all participants are shown in Table 1. In total, 48 of the subjects had type 2 diabetes and 324 did not. The mean ages (SD) of the type 2 diabetes group and the non-diabetic group were 62.6 (9.8) years and 51.8 (11.4) years, respectively. There were significant differences in sex (*p* = 0.032), BMI (*p* = 0.001), number of teeth (*p* < 0.001), ABL (*p* < 0.001), hs-CRP (*p* < 0.001), and periodontal disease severity classification (*p* < 0.001) between the type 2 diabetic group and the non-diabetic group.

Figure 1 shows the distribution of subjects in the nine groups: 119 in group IA; 87 in group IIA; 25 in group IIIA; 33 in group IB; 29 in group IIB; 11 in group IIIB; 26 in group IC; 32 in group IIC; and 10 in group IIIC. The distribution of type 2 diabetes was the highest in group ⅢC with 60.0%, followed by group IIIB with 54.5%, and group IIIA with 28.0% (Table 1).

Multivariate logistic regression analysis with forward selection adjustment for covariates revealed that the presence of type 2 diabetes was significantly associated with age (Odds ratio [OR] = 1.082, 95% confidence interval [CI] = 1.042–1.1124, *p* < 0.001) and BMI (OR = 1.175, 95%CI = 1.061–1.301, *p* = 0.002). Additionally, the presence of type 2 diabetes was significantly associated with periodontal disease severity group ⅢA (OR = 5.108, 95%CI = 1.346–19.381, *p* = 0.017), group ⅢB (OR = 9.626, 95%CI = 1.950–47.528, *p* = 0.005), and group ⅢC (OR = 12.386, 95%CI = 2.464–62.276, *p* = 0.002) when compared with group ⅠA (Table 2). Moreover, the model fitness of the analysis in Table 2 was good (Nagelkerke’s R^2^ = 0.357, *p* < 0.001, the Hosmer–Lemeshow test, χ2 = 9.012, *p* = 0.341).

In the ROC analysis predicting the presence of type 2 diabetes, the AUROC was 0.762 (95%CI = 0.688–0.835, *p* < 0.001) for ABL, and 0.709 (95%CI = 0.635–0.784, *p* < 0.001) for hs-CRP (Figure 2).

## 4. Discussion

This is the first study to show the association between type 2 diabetes and a classification of periodontal disease severity using the combination of ABL and hs-CRP in Japanese people. Furthermore, it was found that ABL, which can be identified by oral radiographs taken at the time of dental treatment, and hs-CRP may be candidates for identifying individuals with asymptomatic type 2 diabetes.

Several cohort studies have reported that periodontal disease affects the onset of diabetes and glycemic control [19,20,21,22,23,24,25,26]. The US National Health and Nutrition Examination Survey showed that the prevalence of diabetes in patients with periodontal disease was approximately twice as high as in patients without periodontal disease [24,25]. In a cohort study in Hisayama Town, it was reported that the prevalence of periodontal disease was significantly higher in Japanese patients who developed impaired glucose tolerance over 10 years than in those who did not [21]. Previous studies have shown that periodontitis is associated with increased risk of diabetic microangiopathy in patients with type 2 diabetes [2]. Moreover, periodontal disease and type 2 diabetes have an interactive relationship, although there is little detailed data on the relationship between the two diseases in Japan. One reason for this gap in the literature is that there has been no unified standard for evaluating periodontal disease.

A consensus report, jointly prepared by the editorial board of the American Academy of Periodontology and the American Society of Cardiology and published simultaneously in the American Journal of Cardiology and the Journal of Periodontology, also provides clinical parameters for further research. It stressed the need for more advanced diagnosis of periodontitis by severity, such as the use of biomarkers and proof of ABL by using radiographs [27]. Therefore, in 2011, the Periodontal Medicine Committee of the Japanese Society of Periodontology established criteria for diagnosing the severity of periodontal disease, which are used as the standard when conducting research. ABL is classified into three stages based on the data of previous studies [28,29]: clinically mild, clinically moderate, and clinically severe. Errors are unlikely to occur in the measurement of ABL; however, it does not reflect the situation when a periodontal pocket has healed. The hs-CRP value, which is a biomarker for inflammation, was defined according to the Hisayama study [30] and consists of three stages: mild inflammation, moderate inflammation, and severe inflammation. Hs-CRP is not a marker specific to periodontal disease; however, it is a highly sensitive marker suited to measuring periodontal disease, which is regarded as a mild chronic inflammation. Previous studies have reported that hs-CRP levels are often high in patients with severe periodontitis and decrease with treatment [29,31]. As shown in Table 1, hs-CRP was associated with type 2 diabetes in this study, which is similar to the previous study [12]. This study is also an important epidemiological study to evaluate whether these classifications are valid.

The main result of this study was that the rate of ABL had a higher AUROC value than hs-CRP, suggesting that individuals who have type 2 diabetes may be identified from the results of ROC analysis. Furthermore, in logistic regression analysis, the IIIA, IIIB, and IIIC groups with an alveolar bone resorption rate of 35% or more were associated with type 2 diabetes, regardless of the hs-CRP value. These findings indicate that the rate of ABL caused by periodontal disease, which is an oral factor, may be closely associated with type 2 diabetes. Therefore, it is possible that the rate of ABL that is easily calculated from X-rays can predict type 2 diabetes with high probability. In the present study a dentist measured the rate of ABL, however we calculated the intra-class correlation coefficient (ICC) assuming that two dentists measured it. The intra-rater reliability was 0.94 ± 0.07 and the inter-rater reliability was 0.89, which ensured reliability.

Probing of periodontal pockets has been considered essential to determine the extent of periodontal tissue destruction. However, in infected and inflamed periodontal tissue, the probing test itself often causes bleeding. There is a risk that oral bacteria will penetrate the bleeding site and induce bacteremia and infective endocarditis. Another concern is that patients may be infected with coronavirus disease 2019 (COVID-19). Dental treatment at close range presents a high risk of infection. Until the risk of COVID-19 is resolved, it may be necessary to diagnose type 2 diabetes with a test that can be performed outside the oral cavity. Our findings help diagnose periodontal disease and identify at-risk patients for type 2 diabetes.

There are some limitations in this study. First, all subjects visited Matsumoto Dental University Hospital, and thus lived in a specific region of Japan; they were not representative of the entire population. Second, we did not show an association between type 2 diabetes and elevated glycated hemoglobin level, elevated low-density lipoprotein cholesterol level, albuminuria, smoking, or elevated blood pressure. Third, the variability in the number of classifications of periodontal disease severity into nine groups should be improved for better analysis. The subjects in this study were not sufficient for the precise analysis. As the confidence interval in multivariate logistic regression analysis was large in this result, we should plan to increase the number of subjects in severe periodontal disease. Fourth, this study was a cross-sectional survey of people who visited a physical examination. It is therefore not known whether their diabetes was under control at the time of the examination, and it is likely that they were a mixed group. However, we believe that the mixture of participants with controlled and uncontrolled diabetes was helpful to validate the usefulness of the new classification in both types of patients. Fifth, for X-ray images, although it would have been ideal to standardize on panoramic radiographs, some subjects were measured by intraoral dental radiographs to reduce exposure for the study. There may be differences in alveolar bone resorption rates. There may also be a possibility that proximal caries may affect the alveolar bone resorption rate. The strength of this study is that ABL measured from X-ray images taken during dental treatment associated with type 2 diabetes with a high probability, and therefore patients can be urged to consult the internal medicine department to receive interventions for lifestyle improvement. We previously reported that alveolar bone resorption was effective as a screening factor for carotid artery calcification [32]. We plan to continue investigating further possible associations of ABL with systemic diseases.

## 5. Conclusions

This study showed the association between type 2 diabetes and a classification of periodontal disease severity using the combination of ABL and hs-CRP in Japanese patients. Furthermore, it was found that ABL, which can be identified by oral radiographs taken at the time of dental treatment, and hs-CRP may be candidates for identifying individuals with underdiagnosed type 2 diabetes.

## Figures and Tables

**Figure 1 ijerph-19-08134-f001:**
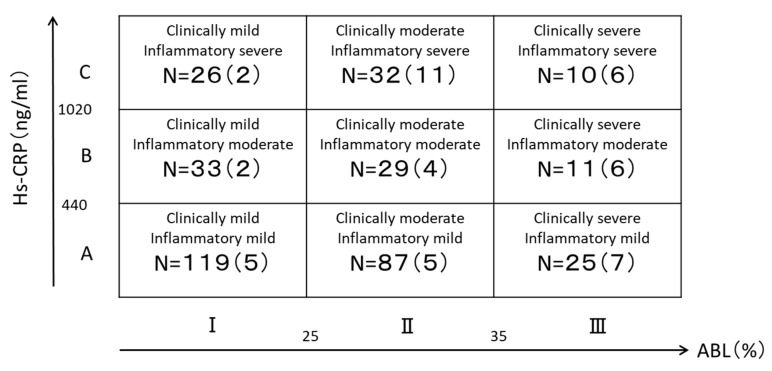
Number of subjects according to classification of periodontal disease severity. ( ): number of patients with type 2 diabetes.

**Figure 2 ijerph-19-08134-f002:**
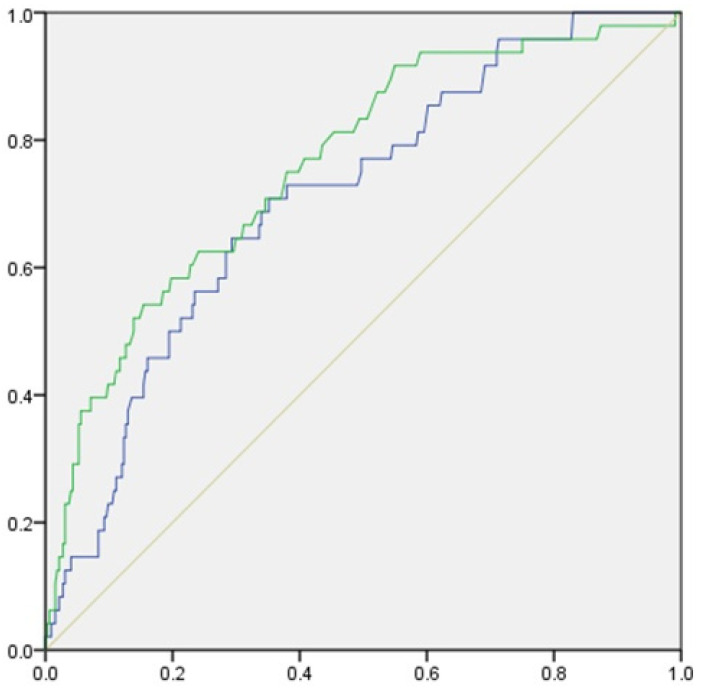
ROC curve of effective factors for screening for type 2 diabetes. The green line represents alveolar bone loss; the blue line represents high-sensitivity C-reactive protein value.

**Table 1 ijerph-19-08134-t001:** The relationship between type 2 diabetes and other variables.

	Type 2 Diabetes (N = 48)	Non-Diabetic (N = 324)	*p*-Value
Age (years)	62.6 ± 9.8	51.8 ± 11.4	<0.001
Sex			
Male	39 (15.5)	213 (84.5)	0.032
Female	9 (7.5)	111 (92.5)	
BMI (kg/m^2^)	25.1 ± 4.2	22.9 ± 3.6	0.001
Smoking			
Yes	12 (17.1)	58 (82.9)	0.239
No	36 (11.9)	266 (88.1)	
Number of teeth	22.2 ± 5.3	25.7 ± 3.9	<0.001
ABL (%)	34.3 ± 11.7	25.6 ± 7.5	<0.001
Ⅰ	9 (5.1)	169 (94.9)	<0.001
Ⅱ	20 (13.5)	128 (86.5)	
Ⅲ	19 (41.3)	27 (58.7)	
Hs-CRP (ng/mL)	1269.4 ± 1690.2	587.5 ± 966.4	<0.001
A	17 (7.4)	214 (92.6)	<0.001
B	12 (16.4)	61 (83.6)	
C	19 (27.9)	49 (72.1)	
Classification of periodontal disease severity			
ⅠA	5 (4.2)	114 (95.8)	<0.001
ⅠB	2 (6.1)	31 (93.9)	
ⅠC	2 (7.7)	24 (92.3)	
ⅡA	5 (5.7)	82 (94.3)	
ⅡB	4 (13.8)	25 (86.2)	
ⅡC	11 (34.4)	21 (65.6)	
ⅢA	7 (28.0)	18 (72.0)	
ⅢB	6 (54.5)	5 (45.5)	
ⅢC	6 (60.0)	4 (40.0)	

N (%) or mean ± SD, BMI—body mass index; ABL—rate of alveolar bone loss; Hs-CRP—high sensitivity C-reactive protein value.

**Table 2 ijerph-19-08134-t002:** Factors associated with type 2 diabetes and the classification of periodontal disease severity—multivariate logistic regression analysis with forward selection.

	Odds Ratio	95%CI	*p*-Value
Age (years)	1.082	1.042–1.124	<0.001
BMI (kg/m^2^)	1.175	1.061–1.301	0.002
Classification of periodontal disease severity			
ⅠA (reference)	1.000		
ⅠB	1.132	0.196–6.535	0.890
ⅠC	0.831	0.141–4.948	0.839
ⅡA	0.955	0.254–3.590	0.946
ⅡB	2.100	0.489–9.023	0.319
ⅡC	3.582	0.988–12.986	0.052
ⅢA	5.108	1.346–19.381	0.017
ⅢB	9.626	1.950–47.528	0.005
ⅢC	12.386	2.464–62.276	0.002

CI—confidence interval; BMI—body mass index.

## Data Availability

Data available on request due to restrictions, e.g., privacy or ethical. The data presented in this study are available on request from the corresponding author.

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
