# Peer review of "Association between Type 2 Diabetes and Classification of Periodontal Disease Severity in Japanese Men and Women: A Cross-Sectional Study"

_ijerph, 2022, doi:10.3390/ijerph19138134_

Round 1
Reviewer 1 Report
A recent paper on the relationship between periodontitis and type 2 diabetes mellitus should be included in the introduction and discussion parts.
Regarding to the authors’ comment on the lack of consensus and uniformity in the definition of periodontitis between the American Association of Periodontology and the European Federation of Periodontology, both of them have been developed in 2012 and 2018 (ref 6 and 7). The sentence “ Therefore, in 2011…” (line 57) should be adjusted.
For the inclusion criteria, patients diagnosed with type 2 diabetes were receiving medication and insulin injection therapy. However, the status of their diabetes was not mentioned. It is important to clarify whether they were controlled type 2 diabetes.
For the exclusion criteria, No. 2 (uncontrolled severe cardiac disease, renal ....) , please mention in details how to justify.

Reviewer 2 Report
The design and concept of the study are not straightforward. A dentist who suspects that the patient may have Diabetes Mellitus and is unaware of it should examine his A1C level and not the CRP level.

Round 2
Reviewer 2 Report
The article now is better understood and is more correct (statistics). The limitation paragraph should include the small number of patients. (lines 249-258)
You also mentioned that occlusal caries might cause bone loss- it should be corrected to proximal caries (lines 256-257)
